# Phenotypic and functional features of B cells from two different human subcutaneous adipose depots

Daniela Frasca[1,2]*, Denisse Garcia[1], Alain Diaz[1], Maria Romero[1], Seth Thaller[3], Bonnie B. Blomberg[1,2]

1 Department of Microbiology and Immunology, University of Miami Miller School of Medicine, Miami, FL, United States of America, 2 Sylvester Comprehensive Cancer Center, University of Miami Miller School of Medicine, Miami, FL, United States of America, 3 Division of Plastic and Reconstructive Surgery, Department of Surgery, University of Miami Miller School of Medicine, Miami, FL, United States of America

* dfrasca@miami.edu

**Data Availability Statement:** All relevant data are within the paper.

**Funding:** AG32576 (DF), AG059719 (DF+BBB), AG023717 (DF+BBB). The funders had no role in

## Abstract

In this study, we have compared frequencies, phenotype, function and metabolic requirements of B cells isolated from the breast and abdominal subcutaneous adipose tissue (AT) of women with obesity who underwent weight reduction surgeries. Results show that B cells from the abdominal AT are more inflammatory than those from the breast, characterized by higher frequencies of inflammatory B cell subsets and higher expression of RNA for inflammatory markers associated with senescence. Secretion of autoimmune antibodies is also higher in the abdominal AT as compared to the breast, and is associated with higher frequencies of autoimmune B cells with the membrane phenotype CD21[low]CD95+ B cells expressing the transcription factor T-bet. Moreover, glucose uptake is higher in B cells from the abdominal AT as compared to the breast, thereby suggesting a better capacity to perform glycolysis, needed to support intrinsic B cell inflammation and autoimmune antibody secretion.

## Introduction

Obesity, defined as a body-mass index (BMI) $\geq 30$ kg/m$^2$, is an inflammatory condition associated with persistent local and systemic low-grade inflammation, similar to what has been observed in older individulas and called inflammaging [1]. Obesity is also associated with chronic immune activation (IA), functional impairment of immune cells, and dysfunctional immunity. Obesity and associated inflammation lead to several debilitating chronic diseases such as type-2 diabetes mellitus [2–4], cardiovascular disease [5, 6], cancer [7], atherosclerosis [8], psoriasis [9], inflammatory bowel disease [10], as well as to increased severity of infectious diseases, such as COVID-19 [11–14].

Obesity is characterized by increased fat mass, mainly due to increased size of the adipocytes, with hypertrophic adipocytes changing their biological function and secretory phenotype, leading to increased secretion of pro-inflammatory cytokines, chemokines and adipokines as well as increased infiltration of immune cells that localize in crown-like

study design, data collection and analysis, decision to publish, or preparation of the manuscript.

**Competing interests:** The authors have declared that no competing interests exist.

structures around the adipocytes. The expansion of adipocyte size leads to remodeling of the adipose tissue (AT), which is characterized by the accumulation of extracellular matrix components occurring concomitantly with the secretion of pro-inflammatory mediators [2]. The human AT includes the subcutaneous and the visceral AT, each with different anatomical depots, structural organization, cellular composition and function, with the visceral being more inflammatory than the subcutaneous AT [15–17].

Previously, we have studied B cells that have infiltrated the human obese subcutaneous AT of women undergoing breast reduction surgeries [18, 19]. We have shown that B cells are recruited by chemokines released by both adipocytes and immune cells (resident and infiltrating), thus generating a positive feedback inflammatory loop responsible for the recruitment of other leukocyte populations to the tissue [18]. We have also reported the presence of pathogenic B cells able to secrete autoimmune antibodies [18, 20]. Although these antibodies can be specific for a variety of molecules, such as nucleic acids, lipids, and proteins [21], we have found that they are specific for adipocyte (AD)-derived proteins, not known as common immunogenic autoantigens. These antibodies are released in large amounts after cell death in the AT under obesity conditions following increased hypoxia and reduced mitochondrial respiration, as well as cell cytotoxicity and DNA damage [18] and have been described to be pathogenic in mouse studies [22]. In our previously published study [18] we have shown that these AD-specific autoimmune antibodies are secreted in the obese AT without any additional stimulation, suggesting an ongoing process of class switch in the tissue leading to B cell differentiation. We have also found that the in vitro incubation of naïve B cells with an adipocyte-conditioned medium which is enriched in class switch cytokines such as IFN-γ and IL-21 (Frasca et al., manuscript submitted) significantly enhances RNA expression of AID, activation-induced cytidine deaminase, the enzyme of class switch recombination and somatic hypermutation. Moreover, the obese AT is enriched in Germinal Center (GC) B cells characterized by intracellular Bcl-6 expression and membrane CD19+CD10+IgD- expression, a phenotype of Dark Zone GC B cells, the centroblast-rich area proximal to the T cell area, known to be the area where class switch occurs [23].

AD-specific autoimmune antibodies are increased in the plasma of elderly lean individuals [24], and we have hypothesized that this may be dependent on the increased deposition of triglycerides on internal tissues and organs (liver, muscle, heart, pancreas, kidney) [25–29] as well as on blood vessels [30].

It is physiologically relevant to elucidate the distribution of these pathogenic B cells in different subcutaneous AT depots and therefore in this study we have compared frequencies, phenotype, function and metabolic requirements of B cells from breast and abdominal subcutaneous AT of women with obesity undergoing breast reduction surgery or panniculectomy (removal of lower abdominal fat) for weight reduction purposes. Results have shown that B cells from the abdominal AT (pannus) are more inflammatory than those from the breast, characterized by higher frequencies of inflammatory B cell subsets and higher expression of RNA for markers of the SASP (senescence-associated secretory phenotype). Secretion of autoimmune antibodies is also higher in the pannus as compared to the breast AT, and is associated with higher glucose uptake and better capacity to perform glycolysis, a characteristic needed to support intrinsic B cell inflammation and autoimmune antibody secretion.

## Materials and methods

### Subjects

We obtained the fresh discarded obese subcutaneous AT from adult female donors undergoing breast reduction surgery [n = 22] or panniculectomy (removal of lower abdominal fat)

**Table 1. Demographic characteristics of enrolled participants.**

|  | BREAST (n = 22) | PANNUS (n = 12) |
| --- | --- | --- |
| Age, mean±SE | 35.91 ± 2.87 | 48.67 ± 3.56 * |
| BMI, mean±SE (range) | 37.37 ± 0.79 (31–43) | 39.92 ± 1.41 (34–49) |
| Race (white/black) | 9/13 | 5/7 |
| Ethnic Categories (Hispanic/Non Hispanic) | 14/8 | 8/4 |

*p<0.05

[n = 12], at the Division of Plastic and Reconstructive Surgery at the University of Miami Hospital. Study participants provided written informed consent. The study was reviewed and approved by the Institutional Review Board (IRB, protocols #20070481 and #20160542), which reviews all human research conducted under the auspices of the University of Miami. All donors were screened for diseases known to alter the immune response or for consumption of medications that could alter the immune response. We excluded subjects with autoimmune, cardiovascular, renal or hepatic diseases, as well as with chronic infectious diseases, cancer, or under substance and/or alcohol abuse. The demographics of the participants are shown in Table 1.

## Isolation of immune cells from the subcutaneous AT

The AT was harvested from surgery patients, weighed and washed with 1X Hanks' Balanced Salt Solution (HBSS). Then, it was resuspended in 1X High Dulbecco's modified Eagle's Medium (DMEM), supplemented with 15 mM HEPES, 1mM Sodium Pyruvate, 100 U/mL Penicillin-Streptomycin, 1% BSA and 200 nM Adenosine). The AT was then minced into small pieces, passed through a 70 μm filter and digested with collagenase type I (SIGMA C-9263) for 1 hour in a 37°C water bath. Digested cells were passed through a 300 μm filter, and centrifuged at 300 g in order to separate the floating adipocytes (AD) from the stromal vascular fraction (SVF) containing the immune cells. Cell pellet (SVF) was resuspended in ACK (Ammonium-Chloride-Potassium) for 3 minutes at room temperature to lyse the Red Blood Cells. SVF was then washed 3 times with DMEM, counted and used in all the experiments.

## Flow cytometry

Cells from the SVF ($5x10^5$) were stained for 20 minutes at room temperature with a Live/Dead detection kit (InVitrogen 1878898), anti-CD45 (Biolegend 368540), anti-CD19 (BD 348794), anti-CD27 (BD 555441) and anti-IgD (BD 555778) antibodies. This protocol allows the identification of the major B cell subsets [naive (IgD+CD27-), IgM memory (IgD+CD27+), switched memory or swIg (IgD-CD27+), and double negative or DN (IgD-CD27-) B cells]. To measure membrane expression of markers associated with autoimmune B cells, cells were also stained with anti-CD95 (Biolegend 305635) and anti-CD21 (Biolegend 354911). Up to $10^5$ events in the lymphocyte gate were acquired on an LSR-Fortessa (BD) and analyzed using FlowJo 10.5.3 software. Single color controls were included in every experiment for compensation. Isotype controls were also used in every experiment to set up the gates.

## PrimeFlow

PrimeFlow RNA assay (ThermoFisher) allows the measurement of intracellular mRNA at single cell level by an amplified Fluorescence In Situ Hybridization technique, in combination with flow cytometry. Unstimulated SVF cells were initially membrane stained with Live/Dead

detection kit and with the following antibodies: anti-CD45, anti-CD19, anti-CD95 and anti-CD21. For T-bet mRNA detection, target probe hybridization was performed using type 1 (AlexaFluor647) probe for T-bet (Affymetrix VA1-16417-06). Negative control was the sample without the target probe. Cells were initially incubated for 2 hours with the probe in a calibrated incubator set to 40˚C and then incubated with the PreAmplification (PreAmp) reagent for 1.5 hours and the Amplification (Amp) reagent for an additional 1.5 hours at 40˚C. After signal amplification, cells were incubated with the label probe at 40˚C for 1 hour. Cells were washed and suspended in staining buffer prior to acquisition. Approximately $10^5$ events were acquired from each sample on an LSR-Fortessa (BD) and analyzed using FlowJo 10.5.3 software. Spectral compensation was completed using single color control samples. Isotype controls were also used in every experiment to set up the gates.

## B cell sorting

SVFs were stained with anti-CD45 and anti-CD19 antibodies. CD19+ B cells were sorted in a Sony SH800 cell sorter. Cell preparations were typically >95% pure.

## RNA extraction and quantitative (q)PCR

After sorting, B cells were resuspended in TRIzol (Ambion) ($10^6$ cells/500 μL), then RNA extracted for quantitative (q)PCR. Total RNA was isolated according to the manufacturer's protocol, eluted into 10 μL distilled water and stored at -80˚C until use.

Reverse Transcriptase (RT) reactions were performed in a Mastercycler Eppendorf Thermocycler to obtain cDNA. Briefly, 2 μL of RNA at the concentration of 0.5 μg/μL were used as template for cDNA synthesis in the RT reaction. Conditions were: 40 minutes at 42˚C and 5 minutes at 65˚C. For miRs quantification, RT reactions were performed in the presence of specific primers. The qPCR reactions were conducted in MicroAmp 96-well plates, and run in the ABI 7300 machine. Calculations were made with ABI software. Briefly, we determined the cycle number at which transcripts reached a significant threshold (Ct). A value for the amount of the target gene, relative to GAPDH for RNAs or to U6 for micro-RNAs (miRs), was calculated and expressed as ΔCt.

Reagents and primers for qPCR amplification, all from ThermoFisher, were the following: GAPDH, Hs99999905_m1; TNF, Hs01113624_g1; IL-6, Hs00985639_m1; IL-8, Hs00174103_m1; CXCR3, Hs01847760_s1; CXCR2, Hs01891184_s1; CCR2, Hs00704702_s1; CCR3, Hs00266213_s1; U6, 001973; miR-155, 002623; miR-16, 000391; miR-181a, 000480; TLR2, Hs02621280_s1; TLR4, Hs00152939_m1; TLR9, Hs00370913_s1; p16$^{INK4}$ (CDKN2A), Hs00923894_m1; p21$^{CIP1/WAF1}$, Hs00355782_m1; Glut1 (SLC2A1), Hs00892681_m1; LDHA, Hs01378790_g1; PDHX, Hs00185790_m1.

## ELISA to measure autoimmune antibodies in culture supernatants

For double strand (ds) DNA-specific and Malondealdehyde (MDA)-specific IgG antibody detection we used the Signosis EA-5002 and MyBioSource MBS390120 kits, respectively.

For adipocyte-specific IgG antibody detection, adipocytes were isolated from the subcutaneous AT of patients undergoing weight reduction surgeries, as previously described [18]. After isolation, the adipocytes were centrifuged in a 5415C Eppendorf microfuge (2,000 rpm, 5 minutes). Total cell lysates were obtained using the M-PER (Mammalian Protein Extraction Reagent, ThermoFisher), according to the manufacturer's instructions. Aliquots of the protein extracts were stored at -80˚C. Protein content was determined by Bradford [31].

Proteins and MDA were used at the concentration of 10 μg/mL in 1xPBS to coat ELISA plates. After 1 hour at room temperature, plates were washed, blocked with 1xPBS containing

1% BSA (washing buffer) and then incubated for 30 minutes at 37˚C. Then samples were added and incubated at room temperature for 3 hours. Wells were washed thoroughly with washing buffer before receiving the detecting antibody goat anti-human IgG-Fc HRP-conjugated (Jackson ImmunoResearch 109-035-008, 1:5000 diluted). After 1 hour incubation at room temperature, wells were washed and given the substrate solution (TMB chromogen; Biosource SB01). Wells were incubated 15–20 minutes at room temperature to allow reactions to develop. Well contents were measured for absorbance at 405 nm.

## Glucose uptake measurements

For the evaluation of glucose uptake, SVFs were stained with the fluorescent glucose analog (2-(N-(7-Nitrobenz-2-oxa-1,3-diazol-4-yl)Amino)-2-Deoxyglucose) (2-NBDG, Thermo Fisher N13195), for 30 minutes at room temperature, at the final concentrations recommended by the manufacturer. SVFs were then washed and stained for additional 20 minutes at room temperature with the Live/Dead detection kit, anti-CD45 and anti-CD19 antibodies. Cells were acquired in a BD LSR Fortessa Flow cytometry instrument (BD), using the FITC channel to detect glucose uptake. Up to $10^5$ events in the lymphocyte gate were acquired and analyzed using FlowJo 10.5.3 software.

## Detection of Reactive Oxygen Species (ROS) using CellROX oxidative stress reagent

SVF cells ($10^6$/mL) were left unstimulated in complete medium [RPMI 1640, supplemented with 10% FBS, 100 U/mL Penicillin-Streptomycin, and 2 mM L-glutamine (all reagents are from Gibco)], at 37˚C, 5% $CO_2$ for 30 minutes and then CellROX® Deep Red Reagent (Thermo Fisher C10422) was added to the wells at a final concentration of 1 μM. After an additional 30 minute incubation, cells were collected and washed with FACS buffer, and stained for 20 minutes at room temperature with the Live/Dead detection kit, anti-CD45 and anti-CD19 antibodies. Cells were washed and later acquired in a BD LSR Fortessa Flow cytometry instrument, using the APC channel to detect the signal from oxidated CellROX® Deep Red Reagent. Up to $10^5$ events in the lymphocyte gate were acquired and analyzed using FlowJo 10.5.3 software.

## Mitostress test

We used a mitostress test to evaluate oxygen consumption rates (OCR), a measure of oxidative phosphorylation, and extracellular acidification rates (ECAR), a measure of glycolysis. The test was conducted in a Seahorse XFp extracellular flux analyzer (Agilent). Briefly, we seeded B cells sorted from pannus or breast SVFs in a plate coated with CellTAK (BD Biosciences). Cells were at the concentration of $2x10^5$/well in XF DMEM medium supplemented with glutamine, glucose and pyruvate (200 μL of each reagent in 20 mL of medium). Maximal respiratory capacity was measured by the addition of the following compounds: Oligomycin (1 μM) to block ATP production, and then FCCP (fluoro-carbonyl cyanide phenylhydrazone, 5 μM), an uncoupling agent, to dissipate proton gradients and allow electron transport and oxygen consumption to operate at maximal rate. Maximal respiration was then suppressed by the addition of Rotenone/Antimycin (1 μM), showing that respiration is mitochondrial.

## Statistical analyses

To examine differences between groups, unpaired Student's t tests (two-tailed) were used. To examine relationships between variables, bivariate Pearson's correlation analyses were performed, using GraphPad Prism version 8.4.3 software, which was used to construct all graphs.

## Results and discussion

### The B cell pool in the pannus is characterized by lower frequencies of naïve and higher higher frequencies of DN B cells as compared to those in the breast

We measured the frequencies of the major B cell subsets after staining of SVFs from breast and pannus donors with anti-CD19, anti-CD27 and anti-IgD antibodies. Naïve B cells are IgD +CD27-, IgM memory are IgD+CD27+, switched memory (swIg) are IgD-CD27+ and DN B cells are IgD-CD27-. Fig 1A shows gating strategies. Fig 1B shows the frequencies of the four B cell subsets in the SVFs from pannus and breast donors, with lower frequencies of naïve and higher frequencies of DN B cells in pannus *versus* breast, and no differences in both IgM memory B cells and swIg B cells. Nevertheless, as we have previously shown [18], we confirm in these AT samples the accumulation of memory B cell subsets (IgM, swIg, DN) *versus* naïve B cells, due to the ongoing process of class switch occurring in the obese AT.

DN B cells represent the most pro-inflammatory B cell subset which is significantly increased in the blood of individuals with inflammatory conditions, such as aging [32–34] and

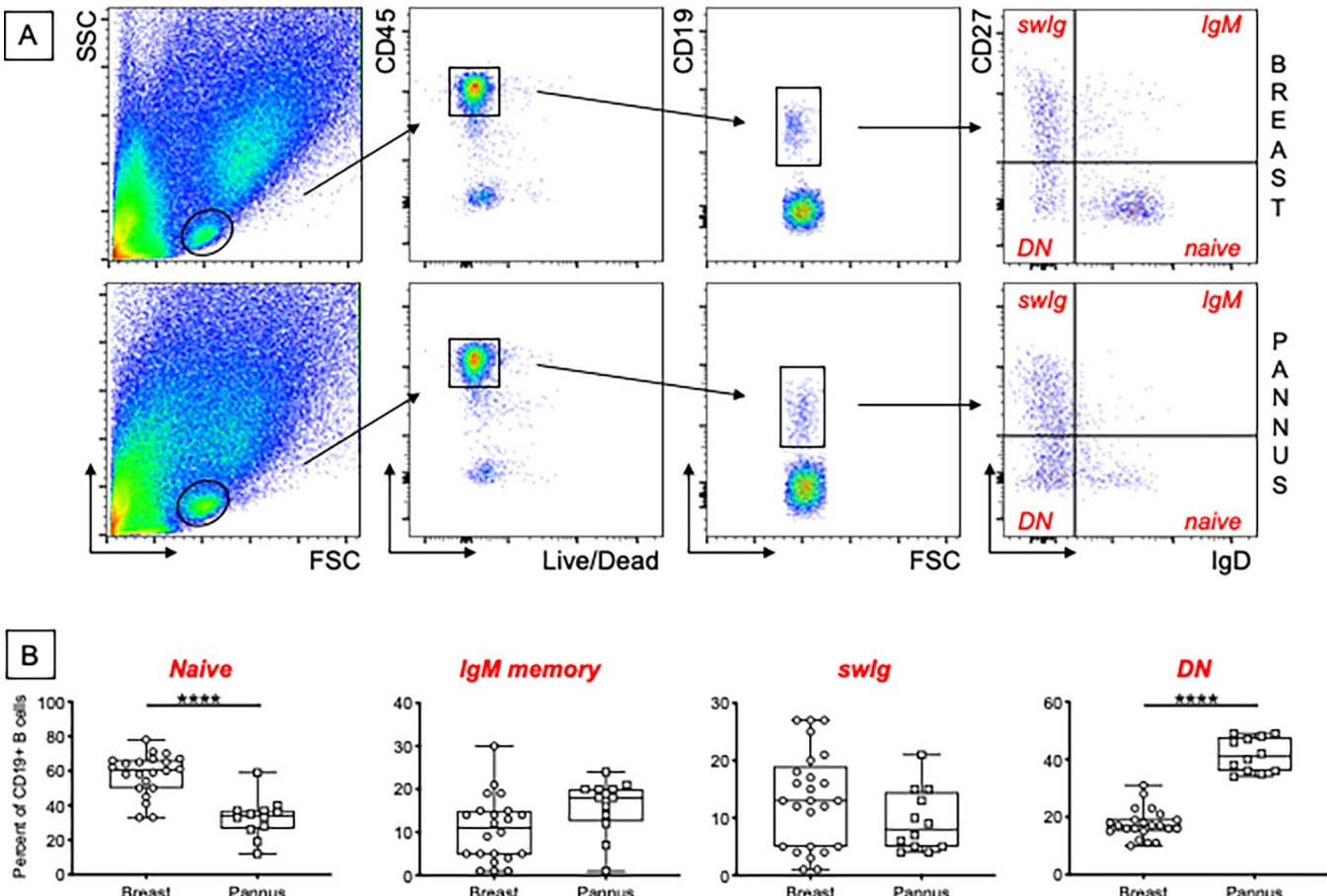

**Fig 1. Lower frequencies of naïve and higher higher frequencies of DN B cells in the B cell pool of pannus *versus* breast.** Unstimulated SVFs were membrane stained with Live/Dead detection kit and anti-CD45/CD19/CD27/IgD antibodies. **A.** Gating strategies and a representative dot plot from one breast (top) and one pannus (bottom) SVF sample to evaluate the major B cell subsets. **B.** Frequencies of naïve, IgM, swIg and DN B cells from all donors (means ±SE), gated on live CD45+ CD19+, are shown. Mean comparisons between groups were performed by Student's t test (two-tailed). ****p<0.0001. Each symbol represents an individual, n = 22 breast donors and n = 12 pannus donors.

obesity [20, 35, 36]. DN B cells are also increased in the blood of patients with autoimmune diseases such as Rheumatoid Arthritis [37], Systemic Lupus Erythematosus [38, 39], Multiple Sclerosis [40], Alzheimer's disease [41, 42], Sjogren's disease [43] and pemphigus [44], as well as of patients with chronic viral [45–47] and parasitic [48] infections. During the recent pandemic, DN B cells have been shown to expand in the blood of COVID-19 patients, with repertoire and functional features similar to those previously described in Lupus patients and associated with symptoms of severe disease [49].

Our results herein confirm and extend our previously published findings that DN B cells expand in the obese AT [18, 20]. The observation that DN B cells expand more in the pannus than in the breast suggests that the pannus AT is more inflammatory than the breast AT.

## Higher frequencies of CD21$^{low}$CD95+ B cells expressing the mRNA for the transcription factor T-bet in pannus *versus* breast B cells

We have previously shown that secretion of IgG antibodies with autoimmune specificities occurs in the AT of individuals with obesity undergoing breast reduction surgeries in the absence of any exogenous stimulation [18]. Therefore, we compared by PrimeFlow frequencies of CD21$^{low}$CD95+ B cells, a phenotype associated with autoimmunity [35, 37], in unstimulated SVFs from breast and pannus donors. The reason to evaluate unstimulated SVFs is because the ongoing process of class switch occurring in the obese AT leads to the secretion of IgG autoimmune antibodies without the need of any additional exogenous stimulation, as we have previously demonstrated [18]. CD21 is the complement receptor for C3d [50], CD95 is Fas ligand [51]. On CD21$^{low}$CD95+ B cells, we then measured mRNA expression of the transcription factor T-bet. T-bet is encoded by the *tbx21* gene, which is known to be associated with the secretion of antibodies with autoimmune specificities in both mice [52–54] and humans [18, 55]. Results in Fig 2 (top) show increased frequencies of CD21$^{low}$CD95+ B cells in pannus *versus* breast SVFs. The mRNA expression of T-bet was also found increased in CD21$^{low}$CD95+ B cells from the pannus *versus* those from the breast Fig 2 (bottom). Negative controls with no probe are shown in the center. This high expression of T-bet RNA, in pannus more than in breast B cells, confirms our previously published findings that B cells in the obese AT are already pre-activated, a status leading to spontaneous secretion of autoimmune antibodies [18, 56].

Not only the frequency of CD21$^{low}$CD95+ B cells was increased in pannus *versus* breast SVFs, but also the expression/cell, as shown in Fig 3, in which we measured mean fluorescence intensity, MFI, of CD21 and CD95 membrane expression.

## Higher secretion of autoimmune antibodies in pannus *versus* breast SVF cultures

After showing that B cells in the obese AT are autoimmune B cells with the phenotype T-bet+-CD21$^{low}$CD95+, and in pannus more than in breast (Fig 3), we wanted to measure secretion of autoimmune IgG in culture supernatants of unstimulated SVF samples. We tested 3 different autoimmune specificities: double strand (ds)DNA, malondehyldehyde (MDA) and adipocyte-derived antigens, which are associated with increased DNA damage (measured by dsDNA) [57], increased oxidative stress and lipid peroxidation (measured by MDA) [58, 59], and increased fat mass [measured by adipocyte (AD)-derived protein antigens released by the tissue] [18]. We have previously shown that the AT of individuals with obesity is enriched in IgG autoimmune antibodies with specificity for AD-derived protein antigens [18], not known as typical autoantigens but released in large amounts without exogenous stimulation, due to chronic cell death in the AT and consequent release of "self" antigens able to induce continous activation of B cells without the need of additional stimulation [56]. Results in Fig 4 show

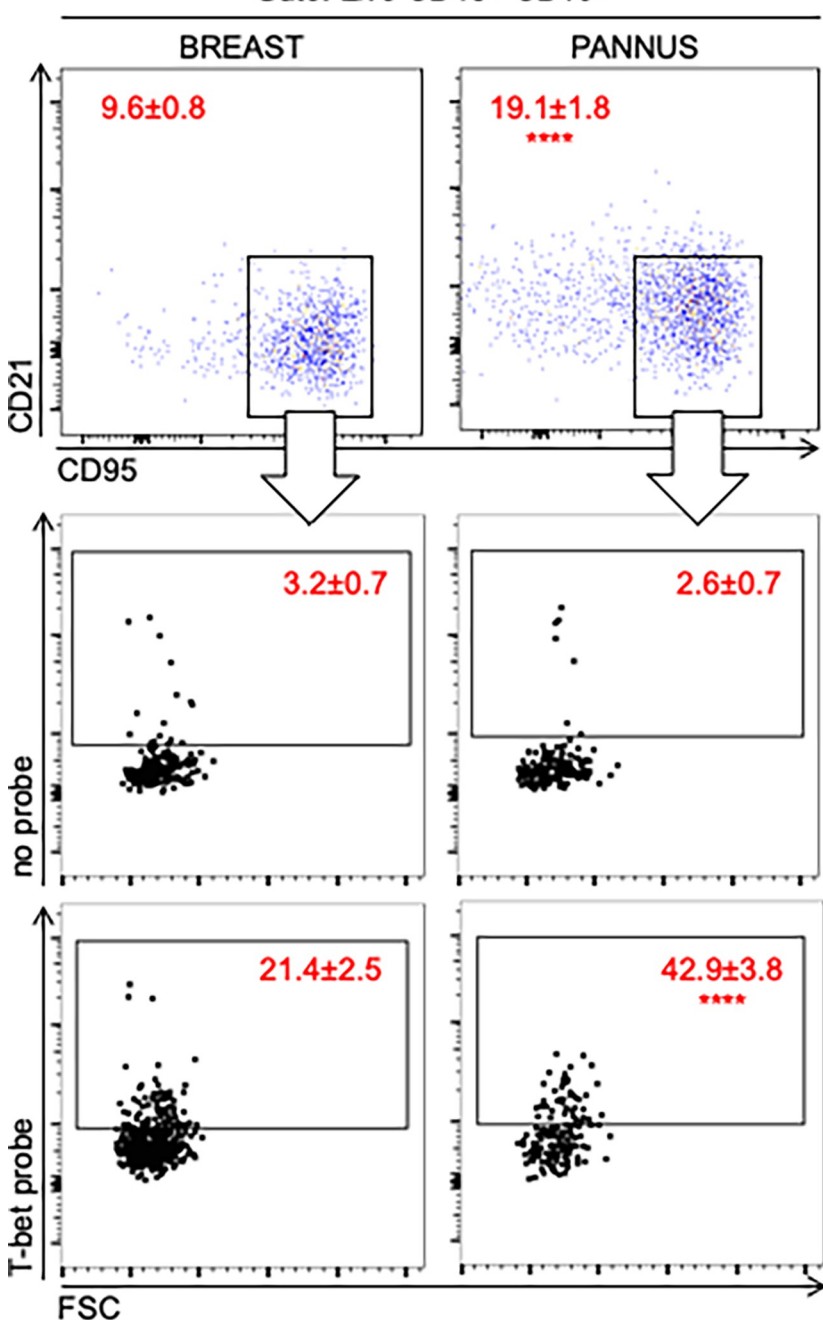

**Fig 2. Higher frequencies of CD21$^{low}$CD95+, T-bet+ B cells in pannus versus breast SVFs.** Unstimulated SVFs were membrane stained with Live/Dead detection kit and anti-CD19/CD21/CD95 antibodies. **Top.** Frequencies of CD21$^{low}$CD95+ B cells (means±SE) in SVFs from breast (n = 10) and pannus (n = 10) donors are shown in red. **Bottom.** Detection of T-bet+ in CD21$^{low}$CD95+ B cells. Means±SE of mRNA expression are shown in red. Negative controls (no probe) are also shown. Mean comparisons between groups were performed by Student's t test (two-tailed). ****p<0.0001.

increased secretion of autoimmune IgG antibodies for these 3 specificities in supernatants of unstimulated SVF samples from the pannus as compared to those from the breast. These results confirm and extend our previously published results that the obese AT is a crucial site

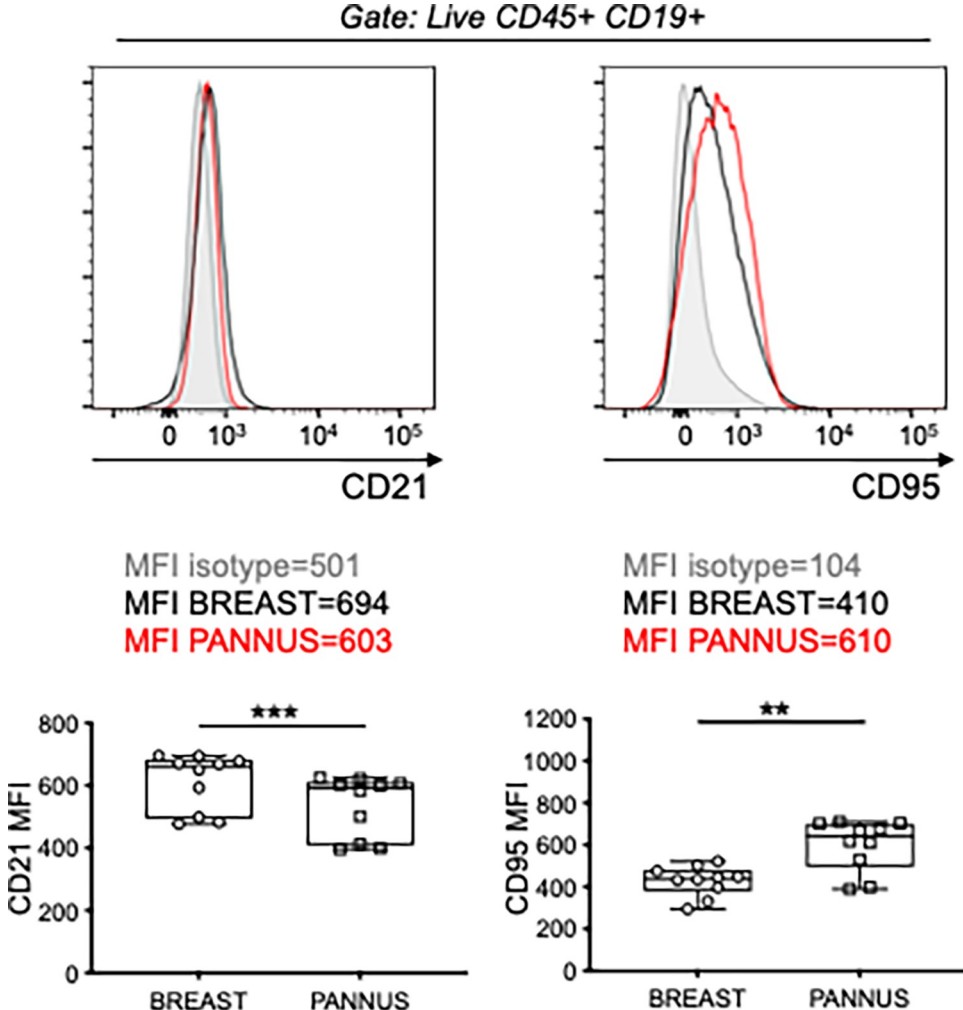

**Fig 3. Increased mean fluorescence intensity of CD95 and CD21 on B cells from pannus *versus* breast SVFs.**
Unstimulated SVFs were stained as in Fig 3. Results show mean fluorescence intensity (MFI)±SE for each marker in B cells from breast and pannus SVF samples, as compared to isotype controls. Mean comparisons between groups were performed by Student's t test (two-tailed). **p<0.01, ****p<0.0001. Each symbol represents an individual, n = 10 breast donors and n = 10 pannus donors.

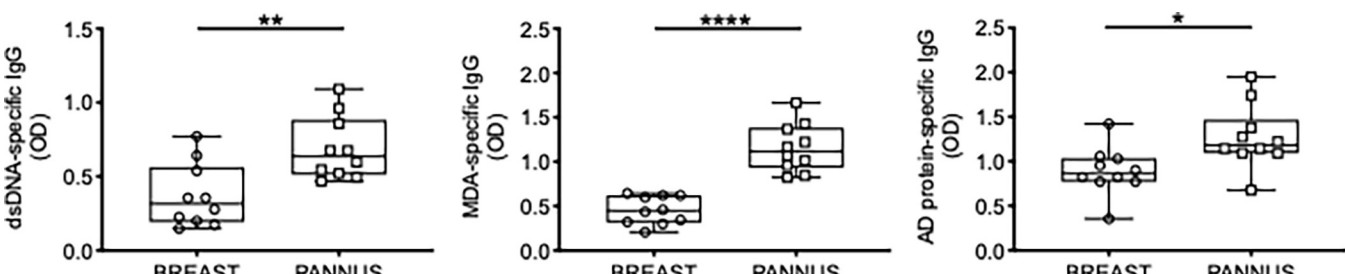

**Fig 4. Higher secretion of IgG antibodies with autoimmune specificities in culture supernatants of pannus *versus* breast SVFs.** Supernatants from unstimulated breast and pannus SVFs were collected after 10 days and analyzed by ELISA for the presence of autoimmune IgG. Mean comparisons between groups were performed by Student's t test (two-tailed). *p<0.05, **p<0.01, ****p<0.0001. Each symbol represents an individual, n = 10 breast donors and n = 10 pannus donors.

for the generation of pathogenic B cells that secrete autoimmune antibodies, as previously demonstrated by mouse studies [22]. These antibodies are pathogenic, as they bind to their target and form immune complexes, inducing potent IA. Moreover, levels of autoantibodies correlate with the severity of the disease and are the most important diagnostic markers for autoimmune diseases. The finding that these autoimmune specificities are also increased in the plasma of healthy elderly individuals [20, 60] has suggested that obesity indeed accelerates age-associated B cell defects.

## Higher expression of RNA for SASP markers in pannus *versus* breast B cells

Pathogenic B cells are pro-inflammatory and express RNA for multiple pro-inflammatory markers of the SASP. Therefore, we evaluated the expression of RNA for SASP markers in B cells sorted from breast and pannus SVF samples. The accumulation of senescent T cells [61] and macrophages [62] in the human obese AT has been reported, but almost nothing is known about the accumulation of senescent B cells. For this series of experiments, B cells were sorted and after sorting were resuspended in TRIzol. The RNA was extracted and the expression of SASP markers evaluated by qPCR. We measured RNA expression of the following pro-inflammatory markers: cytokines (TNF, IL-6), chemokines (IL-8), chemokine receptors (CXCR3, CXCR2, CCR2, CCR3), miRs (miR-155, miR-16, miR-181a), TLRs (TLR2, TLR4, TLR9) and cell cycle inhibitors and markers of proliferation arrest (p16$^{INK4}$, p21$^{CIP1/WAF1}$). Pro-inflammatory cytokines and chemokines, miRs, TLRs and cell inhibitors are all associated with IA and pathogenic B cell responses. Chemokine receptors are associated with the recruitment of B cells to the AT under obesity conditions. Heatmaps in Fig 5 show increased expression of all these markers in pannus *versus* breast B cells.

Our results identify potential molecular and cellular targets of intervention to treat obesity and associated conditions. Several therapeutic protocols to either selectively remove senescent cells and/or suppress the SASP with senolytics have already shown promising results in chronologically old and progeroid mice [63], and in human diseases such as type-2 diabetes mellitus [64]. Because senescent cells accumulate in the AT during aging, leading to production of inflammatory products and ROS, cell death and infiltration of pro-inflammatory immune cells [65, 66], our results also suggest that the selective removal of senescent cells from the AT may represent an effective therapeutic option to extend healthy lifespan and delay the development of inflammatory-based age-associated diseases.

## Higher expression of RNA for metabolic markers in pannus *versus* breast B cells

Senescent cells are metabolically active, a condition necessary to support the SASP, and engage in robust metabolic reprogramming to generate sufficient energy to fuel their demands to support autoantibody secretion. We measured the uptake of glucose by flow cytometry, using the glucose analog (2-(N-(7-Nitrobenz-2-oxa-1,3-diazol-4-yl)Amino)-2-Deoxyglucose) (2-NBDG). Results show increased glucose uptake (Fig 6A and 6B) in B cells from the pannus as compared to those from the breast. Results in Fig 6C also show increased RNA expression of the glucose transporter Glut1 in B cells sorted from breast and pannus SVF samples. The importance of Glut1 in the metabolic reprogramming of cells undergoing proliferation and antibody production has been shown in studies in which its deletion significantly decreases B cell proliferation and antibody secretion [67, 68].

The inflammatory microenvironment of the AT under obesity conditions generates signals that induce a metabolic switch in both myeloid (monocytes and macrophages) and lymphoid (T, B, NK cells) cells, leading to the up-regulation of glucose uptake and consequent activation

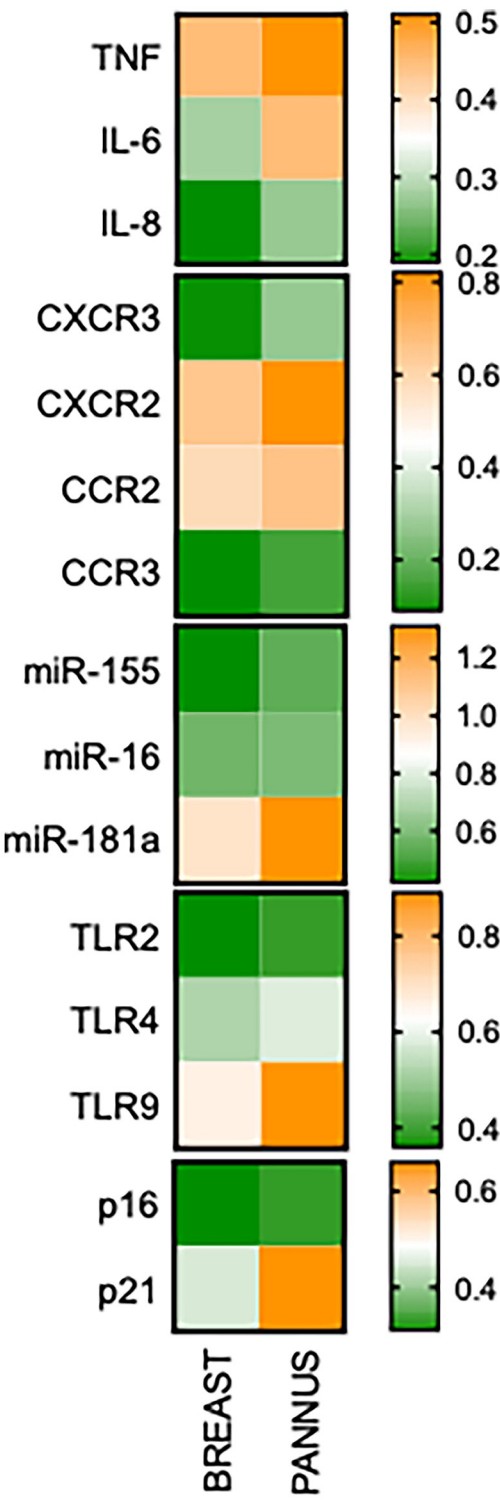

**Fig 5. B cells sorted from the pannus, as compared to those sorted from the breast, express higher levels of RNA for markers of the SASP.** B cells were isolated from pannus (n = 5) and breast (n = 5) SVFs using flow cytometry and cell sorting. After sorting, B cells were left unstimulated and resuspended in TRIzol. The RNA was then extracted and the expression of SASP markers evaluated by qPCR. Heatmap shows qPCR measures of RNA expression of target genes, relative to the housekeeping genes GAPDH or U6 (for miRs quantification), calculated as $2^{-\Delta Cts}$.

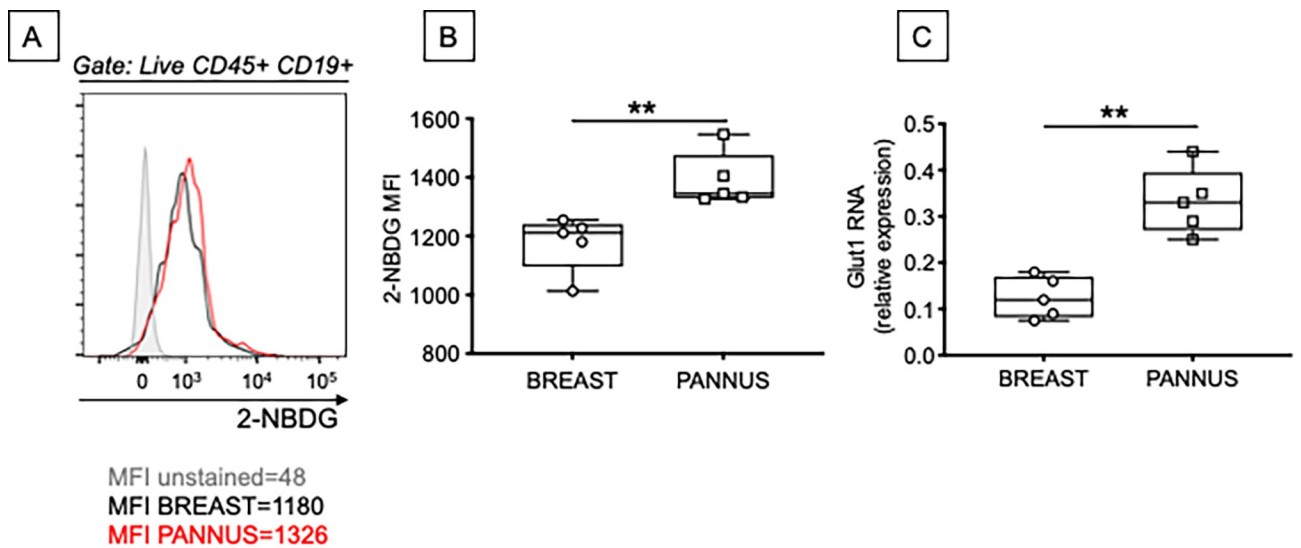

**Fig 6. B cells sorted from the pannus, as compared to those sorted from the breast, express higher levels of 2-NBDG and of the RNA for Glut1.** B cells were isolated from pannus and breast SVFs using flow cytometry and cell sorting. After sorting, B cells were left unstimulated and divided in two aliquots, one evaluated by flow cytometry for 2-NBDG expression, and another resuspended in TRIzol for RNA expression of Glut1. **A.** Results show MFI of 2-NBDG staining in B cells sorted from pannus and breast SVFs from one representative experiment, as compared to the unstained control. **B.** MFI±SE of 2-NBDG staining from all samples. **C.** Results show qPCR values of RNA expression of Glut1, relative to GAPDH, calculated as $2^{-\Delta Cts}$ in the same B cells in A. Mean comparisons between groups were performed by Student's t test (two-tailed). **p<0.01. Each symbol represents an individual, n = 5 breast donors and n = 5 pannus donors.

of pathways of glucose metabolism. We measured RNA expression of genes encoding enzymes associated with glucose metabolism, in particular pyruvate dehydrogenase (PDHX) and lactate dehydrogenase A (LDHA), in B cells sorted from breast and pannus SVF samples. PDHX is a component of the pyruvate dehydrogenase complex that converts pyruvate into acetyl-CoA

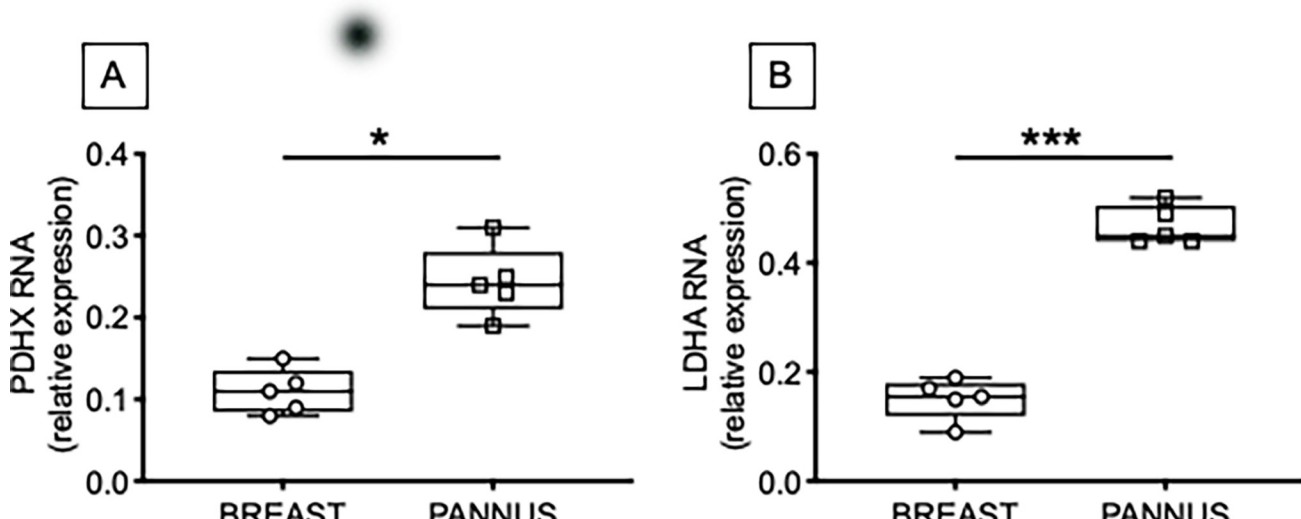

**Fig 7. B cells sorted from the pannus, as compared to those sorted from the breast, express higher levels of RNA for PDHX and LDHA.** B cells were isolated from pannus and breast SVFs using flow cytometry and cell sorting. After sorting, B cells were left unstimulated and resuspended in TRIzol for RNA expression of PDHX (**A**) and LDHA (**B**). Results show qPCR values of RNA expression of PDHX and LDHA, relative to GAPDH, calculated as $2^{-\Delta Cts}$. Mean comparisons between groups were performed by Student's t test (two-tailed). *p<0.05, ***p<0.001. Each symbol represents an individual, n = 5 breast donors and n = 5 pannus donors.

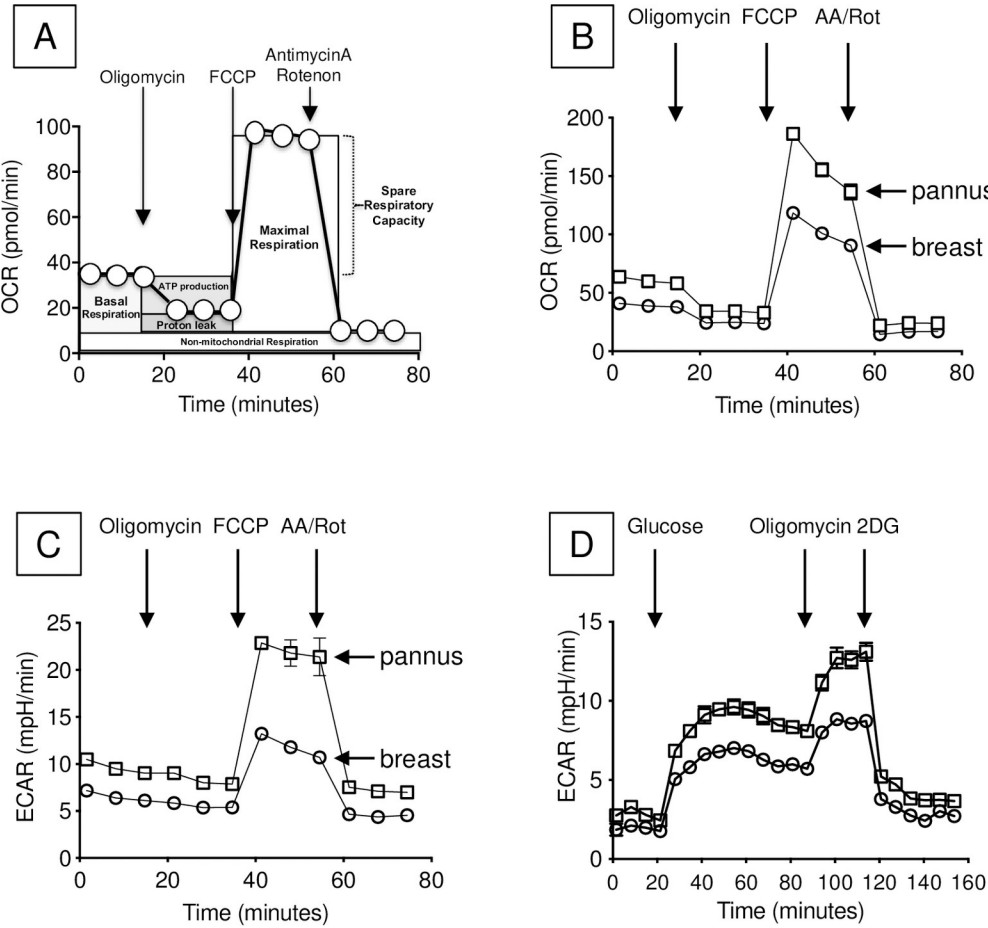

**Fig 8. B cells sorted from the pannus, as compared to those sorted from the breast, are characterized by higher OCR and ECAR.** B cells were isolated from pannus (n = 3) and breast (n = 3) SVFs using flow cytometry and cell sorting. After sorting, B cells were seeded into the CellTAK-coated wells of an extracellular flux analyzer at the concentration of 2x10⁵/well in complete DMEM medium. **A.** Schematic of OCR measured by a mitostress test done in a Seahorse XFp extracellular flux analyzer. **B.** OCR. **C.** ECAR. **D.** Glycolytic test.

and is a measure of oxidative phosphorylation and mitochondrial respiration [69, 70], whereas LDHA converts pyruvate into lactate and is a measure of glycolysis [71–73]. LDHA has high affinity for pyruvate which is preferentially converted into lactate. Results in Fig 7 show increased expression of PDHX (A), and even more of LDHA (B), in B cells from pannus *versus* breast SVFs, suggesting that cells in the pannus microenvironment are more metabolically active than those present in the breast. LDHA has indeed been shown to be transcriptional activated by hypoxia, with hypoxia-inducible factors binding hypoxia-responsive elements in the LDHA promoter [74].

Results in Fig 8 were confirmed using a mitostress test and Seahorse, a technology that allows real-time evaluation of changes in OCR and ECAR, measures of oxidative phosphorylation and of glycolysis, respectively, and calculates a variety of measures of mitochondrial function (basal respiration, maximal respiration, spare respiratory capacity, ATP production, proton leak) with a relatively high throughput. Fig 8 shows, as expected based on the results above, higher OCR and ECAR in B cells sorted from the pannus, as compared to those sorted from the breast. To further support these findings, we have also run a glycolytic test in Seahorse injecting Glucose

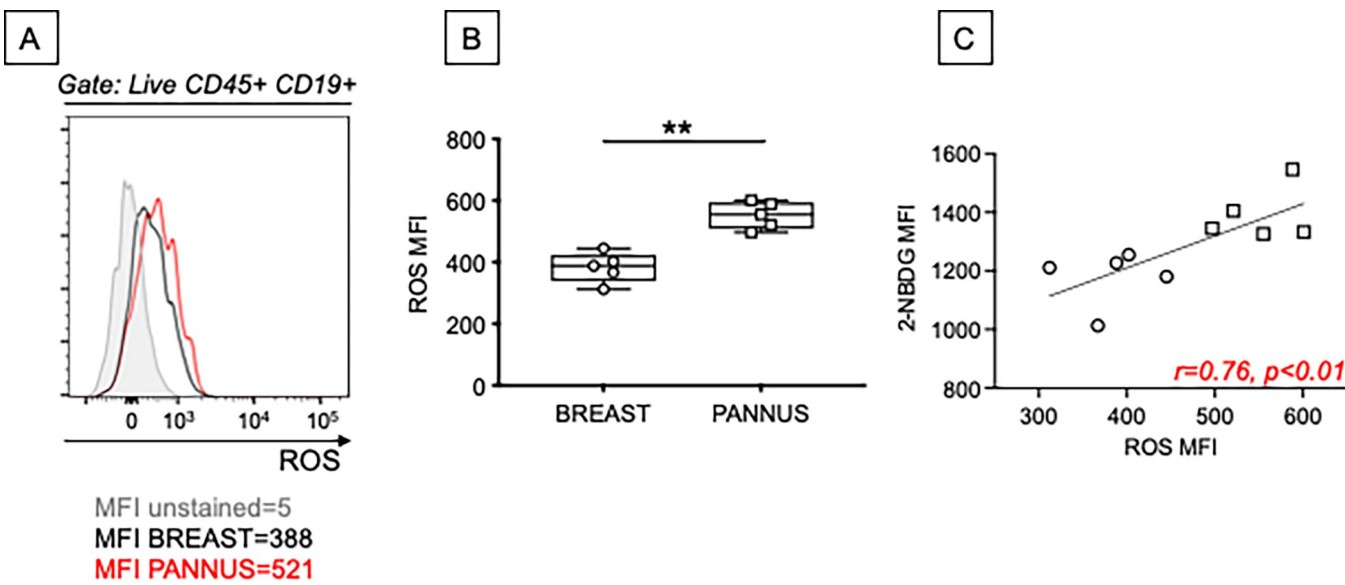

**Fig 9. B cells sorted from the pannus, as compared to those sorted from the breast, express higher levels of ROS and 2-NBDG.** B cells were isolated from pannus and breast SVFs from the same individuals in Figs 7–9. After sorting, B cells were left unstimulated and evaluated by flow cytometry for ROS production. **A.** Results show MFI of ROS staining in B cells sorted from pannus and breast SVFs from one representative experiment, as compared to the unstained control. **B.** MFI±SE of ROS staining from all samples. Mean comparisons between groups were performed by Student's t test (two-tailed). ***p<0.001. **C.** Correlation 2-NBDG and ROS. Each symbol represents an individual, n = 5 breast donors and n = 5 pannus donors.

(10 mM), oligomycin (1 μM) and 2-DG (20 mM), and results have confirmed a highly glycolytic phenotype of B cells from the pannus as compared to those from the breast.

## Higher production of ROS in pannus *versus* breast B cells

Autoimmune antibody secretion is accompanied by metabolic changes associated with the development of pro-inflammatory processes and oxidative stress. Increased levels of glucose in the extracellular microenvironment has been shown to drive the metabolic reprogramming that leads to the differentiation of autoreactive T cells with concomitant production of pro-inflammatory cytokines, ROS and ROS-induced lipid peroxidation [75]. We wanted to know if the differentiation of autoreactive B cells involves similar pathways. We measured ROS levels in B cells from pannus and breast SVFs. Results (Fig 9A and 9B) show increased ROS production in B cells from pannus *versus* breast SVFs. As expected, 2-NBDG and ROS levels were significantly and positively correlated (Fig 9C).

## Conclusions

Several published findings obtained in experimental animals and humans have clearly indicated differences between B cells in the VAT and the SAT, with the VAT being highly enriched in subsets that are more inflammatory and pathogenic. However, almost nothing is known about differences between B cells in different SAT depots. Our results herein, comparing frequencies, phenotype, function and metabolic requirements of B cells from the breast and the abdominal subcutaneous AT of obese female donors, have shown significant differences between the two distinct anatomical depots. B cells from the pannus are more inflammatory and pathogenic than those from the breast, and characterized by a hyper-metabolic phenotype, needed to support their function. We have very preliminary data on B cells from the breast and the abdominal subcutaneous AT of obese male donors showing results comparable to

those herein. Although our results need to be further extended to other AT depots, they are among the first obtained with human AT samples. Our results suggest that B cell depletion may be an effective immunotherapeutic intervention to ameliorate a long list of obesity-associated conditions and co-morbidities. Although the use of anti-CD20 depleting antibodies (Rituximab) have shown positive clinical responses in patients with autoimmune diseases, this treatment is not specific for pathogenic B cell subsets and protective B cell subsets are also drastically reduced by this treatment. Therefore, future studies must focus on the design of therapies that precisely target pathogenic B cells.

## Acknowledgments

We thank nurses and staff of the Ambulatory Surgery. We are deeply thankful to all individuals who have agreed to participate in this study donating their discarded surgery samples.

## Author Contributions

**Conceptualization:** Daniela Frasca.

**Data curation:** Daniela Frasca, Denisse Garcia, Alain Diaz, Maria Romero.

**Formal analysis:** Daniela Frasca.

**Funding acquisition:** Daniela Frasca, Bonnie B. Blomberg.

**Investigation:** Daniela Frasca, Denisse Garcia, Alain Diaz, Maria Romero.

**Methodology:** Daniela Frasca, Seth Thaller.

**Project administration:** Daniela Frasca.

**Resources:** Daniela Frasca, Seth Thaller.

**Supervision:** Daniela Frasca.

**Validation:** Daniela Frasca.

**Writing – original draft:** Daniela Frasca.

**Writing – review & editing:** Denisse Garcia, Alain Diaz, Maria Romero, Seth Thaller, Bonnie B. Blomberg.

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
