## [Decision Letter · Decision Letter 0]

2 Oct 2022

PONE-D-22-22884Phenotypic and functional features of B cells from two different human subcutaneous adipose depotsPLOS ONE

Dear Dr. Frasca,

Thank you for submitting your manuscript to PLOS ONE. After careful consideration, we feel that it has merit but does not fully meet PLOS ONE’s publication criteria as it currently stands. Therefore, we invite you to submit a revised version of the manuscript that addresses the points raised during the review process. Your manuscript was reviewed by two knowledgeable referees in this area and their comments are appended. As you will see they both had several concerns that will need to be addressed by the authors before I can proceed further. In particularly, the reviewer#1 pointed out several issues related to experimental setting and interpretations of the results. The authors need to properly address/respond to all these issues to satisfy the reviewers. Please submit your revised manuscript by Nov 16 2022 11:59PM. If you will need more time than this to complete your revisions, please reply to this message or contact the journal office at plosone@plos.org. Please include the following items when submitting your revised manuscript:A rebuttal letter that responds to each point raised by the academic editor and reviewer(s). You should upload this letter as a separate file labeled 'Response to Reviewers'.A marked-up copy of your manuscript that highlights changes made to the original version. You should upload this as a separate file labeled 'Revised Manuscript with Track Changes'.An unmarked version of your revised paper without tracked changes. You should upload this as a separate file labeled 'Manuscript'.

We look forward to receiving your revised manuscript.

Kind regards,

Makoto Kanzaki, Ph.D.

Academic Editor

PLOS ONE

Journal Requirements:

Frasca, D., Diaz, A., Romero, M. and Blomberg, B.B., 2021. Phenotypic and functional characterization of Double Negative B cells in the blood of individuals with obesity. Frontiers in immunology, 12, p.616650. https://doi.org/10.3389/fimmu.2021.616650

Frasca, D., Romero, M., Diaz, A., Garcia, D., Thaller, S. and Blomberg, B.B., 2021. B Cells with a Senescent-Associated Secretory Phenotype Accumulate in the Adipose Tissue of Individuals with Obesity. International journal of molecular sciences, 22(4), p.1839. https://doi.org/10.3390/ijms22041839

In your revision ensure you cite all your sources (including your own works), and quote or rephrase any duplicated text outside the methods section. Further consideration is dependent on these concerns being addressed.

"AG32576 (DF), AG059719 (DF+BBB), AG023717 (DF+BBB)"

"AG32576 (DF), AG059719 (DF+BBB), AG023717 (DF+BBB)"

"This study is supported by NIH awards AG32576, AG059719, AG023717. We thank nurses and staff of the Ambulatory Surgery. We are deeply thankful to all individuals who have agreed to participate in this study donating their discarded surgery samples."

"The authors declare no competing interests"

6. PLOS requires an ORCID iD for the corresponding author in Editorial Manager on papers submitted after December 6th, 2016. Please ensure that you have an ORCID iD and that it is validated in Editorial Manager. To do this, go to ‘Update my Information’ (in the upper left-hand corner of the main menu), and click on the Fetch/Validate link next to the ORCID field. This will take you to the ORCID site and allow you to create a new iD or authenticate a pre-existing iD in Editorial Manager. Please see the following video for instructions on linking an ORCID iD to your Editorial Manager account: https://www.youtube.com/watch?v=_xcclfuvtxQ.

Reviewers' comments:

Reviewer's Responses to Questions

**Comments to the Author**

1. Is the manuscript technically sound, and do the data support the conclusions?

Reviewer #1: Partly

Reviewer #2: Yes

2. Has the statistical analysis been performed appropriately and rigorously? 

Reviewer #1: Yes

Reviewer #2: Yes

3. Have the authors made all data underlying the findings in their manuscript fully available?

Reviewer #1: Yes

Reviewer #2: Yes

4. Is the manuscript presented in an intelligible fashion and written in standard English?

Reviewer #1: Yes

Reviewer #2: Yes

5. Review Comments to the Author

Reviewer #1: In this manuscript, the authors build on their previous body of work characterizing the phenotype and metabolic changes of B cells in two different depots of human adipose tissue, the breast and the pannus. Interestingly the authors note that the B cells in the pannus contain more DN B cells, are more activated, secrete more autoantibodies, include more senescent markers, and are metabolically different than the B cells in the breast adipose. This is a compelling comparison in primary human adipose tissue from obese female patients and will be of interest to a broad audience in the field.

While the conclusion is intriguing and consistent with previous work from the other investigators and the authors themselves, there are some critical flaws in the data presentation and interpretation which must be corrected to validate the data. Those concerns are outlined here:

In Figure 2, the authors purport to capture CD21lo cells with their CD95/CD21lo gate. However, they are in fact capturing all CD95+ cells regardless of CD21 level, so they are not studying the intended population (CD95+CD21lo). To capture CD21lo cells, the gate must stop at a CD21 MFI of less than the Max, perhaps 500 or so.

In Figure 3, the unstained sample in the first left panel is described as having an MFI of 8804. The X-axis has no labels, so this is impossible to confirm, but it looks like 1 or 2 logs, not 8 logs. More importantly, the MFI of the unstained sample is 40 times higher than the MFI of either of the experimental samples: Breast, 220 and Pannus, 124. Unless I am missing something, any perceived differences in the MFIs of CD21 in the experimental sample are meaningless if they are all actually below the background of unstained samples. This makes the data in the lower left panel completely invalid.

Figure 6, 7: labeling the Y axis as PCR “values” is too vague. Please be more specific.

Figure 9: The X axis for the histograms is unlabeled, but the average MFIs, noted to be 1227 and 1405, both fall before the first log decade (1000?) in the histogram, which is confusing. Please label the X axis with numbers.

Reviewer #2: in the manuscript by Daniela Frasca and collaborators, the authors demonstrate how B cells from abdominal fat are metabolically more active than B cells from breast, which support increased B cell inflammation and autoimmune antibody secretion. These findings support the idea that visceral fat is more pathogenic than other fat pads, which has been observed by others. Because B cells tend to be less investigated, especially in humans, this manuscript is of relevance in the field of obesity-mediated inflammation. Despite the manuscript is well written and presented, there are several minor points that need to be addressed. Additionally, the discussion section needs to be extended.

Minor points

1. Abstract: Authors only measure glucose uptake. Line 8 in the abstract needs to be rephrased. It says “nutrient uptake” which can involve also amino acids or other nutrients that have not been measured.

2. Intro line 2: needs to be rephrased, as sentence sounds like a BMI greater than 30 associates with inflammaging, but that will only occur (or at least expected to occur) in older individuals but not in younger ones. Especially in this work where cohorts are in the middle-aged range.

3. In the second paragraph of the introduction is indicated that increased fat mass during obesity is mainly due to increased size of the adipocytes. It’d be more accurate if authors briefly mention that changes in adipocyte quality are also driving pathogenic changes in their secretome.

4. I think it’d be clearer if reference 18 and 19 are cited right after the sentence that talks about the details of the findings.

5. Typo on the 1st sentence of the last paragraph in the introduction “delucidate”

6. Line 9 in the last paragraph of the introduction mentions that the secretion of autoantibodies associates with higher nutrient uptake. Since only glucose uptake has been measured, authors should consider switching “nutrient” for “glucose”

7. Include a table with demographic info of the patients and indicate if both groups are BMI matched

8. Include if DMEM was supplemented for SVF isolation

9. Include information of RMPI supplementation for cell culture

10. Page 7 results and discussion: in line 3 authors should include markers that define DN cells. In line 4: introduce why authors look at IgM and swgl markers to make it easier for non-immunologist readers

11. Page 8 line 8, add reference

12. Explain why cells are analyzed without any stimulation. That will help to better understand the concept of “pre-activated” used in the last paragraph of page 8.

13. Figure 1 legend: add in brackets the N’s for each group analyzed

Mayor points

1. The panniculectomy female cohort age average is 48yo, while the breast reduction surgery group average is 36yo. Discussion should address the possibility that B cells from breast fat are less pro-inflammatory due to changes in overall resident/infiltrating immune cells mediated by pre- and menopausal states (like increased Treg). It is known that those states associate with changes in leptin and changes in fat redistribution. Due to the role of leptin in the activation of B cells, it is important either to add a table with the information of these menopausal states and/or possible therapy hormones from each donor, or address/discuss the possibility of differences mediated by those changes in age range.

2. It would be appropriate to include whether donors with type 2 diabetes have been excluded, to avoid anti-inflammatory effects of glycemic control drugs such as metformin, considering they are all in the obese category and there are chances of donors being either pre-diabetic or diabetic.

3. Include in the discussion whether these changes are predicted or not in males

4. Add in the discussion if there are any known changes in the frequency of DN B cells between males and females (mice and/or humans)

5. Extend the discussed part about the potential therapeutic effect of depleting B cells versus depleting just DN B cells.

6. PLOS authors have the option to publish the peer review history of their article (what does this mean?). If published, this will include your full peer review and any attached files.

Reviewer #1: No

Reviewer #2: No

---

## [Author Response · Author response to Decision Letter 0]

15 Dec 2022

Reviewer #1

In this manuscript, the authors build on their previous body of work characterizing the phenotype and metabolic changes of B cells in two different depots of human adipose tissue, the breast and the pannus. Interestingly the authors note that the B cells in the pannus contain more DN B cells, are more activated, secrete more autoantibodies, include more senescent markers, and are metabolically different than the B cells in the breast adipose. This is a compelling comparison in primary human adipose tissue from obese female patients and will be of interest to a broad audience in the field.

Thank you for the positive feedback!

While the conclusion is intriguing and consistent with previous work from the other investigators and the authors themselves, there are some critical flaws in the data presentation and interpretation which must be corrected to validate the data. Those concerns are outlined here:

In Figure 2, the authors purport to capture CD21lo cells with their CD95/CD21lo gate. However, they are in fact capturing all CD95+ cells regardless of CD21 level, so they are not studying the intended population (CD95+CD21lo). To capture CD21lo cells, the gate must stop at a CD21 MFI of less than the Max, perhaps 500 or so.

The reviewer is absolutely correct and we apologize for the mistake, now fixed changing the gate (Fig. 2new).

In Figure 3, the unstained sample in the first left panel is described as having an MFI of 8804. The X-axis has no labels, so this is impossible to confirm, but it looks like 1 or 2 logs, not 8 logs. More importantly, the MFI of the unstained sample is 40 times higher than the MFI of either of the experimental samples: Breast, 220 and Pannus, 124. Unless I am missing something, any perceived differences in the MFIs of CD21 in the experimental sample are meaningless if they are all actually below the background of unstained samples. This makes the data in the lower left panel completely invalid.

The reviewer is correct again and we apologize for the mistake, please see Fig. 3new in which we have re-analyzed all the samples.

Figure 6, 7: labeling the Y axis as PCR “values” is too vague. Please be more specific.

We have changed the label of the Y axes, as suggested by the reviewer. We have also changed the legends of these figures to better explain what we have measured. 

As to Fig. 6, adding the X axis label (the reviewer suggested the same for Fig. 9), we realized that there was a mistake in Fig. 6A and B which is now fixed. We were reporting ROS data, not 2-NBDG. We apologize. 

Figure 9: The X axis for the histograms is unlabeled, but the average MFIs, noted to be 1227 and 1405, both fall before the first log decade (1000?) in the histogram, which is confusing. Please label the X axis with numbers.

We have added the X axis label, as suggested, and we have also fixed the previous mistake (showing 2-NBDG rather than ROS data). Thank you for making us aware of the problem with the axis. Again, sorry for having made all these mistakes.

Reviewer #2

In the manuscript by Daniela Frasca and collaborators, the authors demonstrate how B cells from abdominal fat are metabolically more active than B cells from breast, which support increased B cell inflammation and autoimmune antibody secretion. These findings support the idea that visceral fat is more pathogenic than other fat pads, which has been observed by others. Because B cells tend to be less investigated, especially in humans, this manuscript is of relevance in the field of obesity-mediated inflammation. Despite the manuscript is well written and presented, there are several minor points that need to be addressed. Additionally, the discussion section needs to be extended.

Thank you very much for the positive feedback!

Minor points

1. Abstract: Authors only measure glucose uptake. Line 8 in the abstract needs to be rephrased. It says “nutrient uptake” which can involve also amino acids or other nutrients that have not been measured.

The reviewer is correct. We changed “nutrient uptake” with “glucose uptake”.

2. Intro line 2: needs to be rephrased, as sentence sounds like a BMI greater than 30 associates with inflammaging, but that will only occur (or at least expected to occur) in older individuals but not in younger ones. Especially in this work where cohorts are in the middle-aged range.

We have rephrased the sentence as suggested and we hope that the reviewer agrees with it.

3. In the second paragraph of the introduction is indicated that increased fat mass during obesity is mainly due to increased size of the adipocytes. It’d be more accurate if authors briefly mention that changes in adipocyte quality are also driving pathogenic changes in their secretome.

Correct. We have added a clarification.

4. I think it’d be clearer if reference 18 and 19 are cited right after the sentence that talks about the details of the findings.

Done.

5. Typo on the 1st sentence of the last paragraph in the introduction “delucidate”

This has now been corrected. Thank you!

6. Line 9 in the last paragraph of the introduction mentions that the secretion of autoantibodies associates with higher nutrient uptake. Since only glucose uptake has been measured, authors should consider switching “nutrient” for “glucose”

Done. We replaced the word “nutrient” with “glucose”. 

7. Include a table with demographic info of the patients and indicate if both groups are BMI matched.

Done. Table 1 is at the end of the paragraph “Subjects” in Materials and methods.

8. Include if DMEM was supplemented for SVF isolation.

Done. It is in the paragraph “Isolation of immune cells from the subcutaneous AT” in Materials and methods.

9. Include information of RMPI supplementation for cell culture

Done. It is in the paragraph “Detection of Reactive Oxygen Species (ROS) using CellROX Oxidative Stress Reagent” in Materials and methods.

10. Page 7 results and discussion: in line 3 authors should include markers that define DN cells. 

In line 4: introduce why authors look at IgM and swgl markers to make it easier for non-immunologist readers

Done. It is in the first of paragraph “The B cell pool in the pannus is characterized by lower frequencies of naïve and higher higher frequencies of DN B cells as compared to those in the breast” in Results.

11. Page 8 line 8, add reference

Done.

12. Explain why cells are analyzed without any stimulation. That will help to better understand the concept of “pre-activated” used in the last paragraph of page 8.

Done. It is in the paragraph “Higher frequencies of CD21lowCD95+ B cells expressing the mRNA for the transcription factor T-bet in pannus versus breast B cells ” in Results.

13. Figure 1 legend: add in brackets the N’s for each group analyzed.

This request is not clear to me. In each figure, each symbol represents an individual. For this reason we have not indicated the number of individuals in each experiment. Nevertheless, a clarification and the numbers have now been included in each figure legend.

Mayor points

1. The panniculectomy female cohort age average is 48yo, while the breast reduction surgery group average is 36yo. Discussion should address the possibility that B cells from breast fat are less pro-inflammatory due to changes in overall resident/infiltrating immune cells mediated by pre- and menopausal states (like increased Treg). It is known that those states associate with changes in leptin and changes in fat redistribution. Due to the role of leptin in the activation of B cells, it is important either to add a table with the information of these menopausal states and/or possible therapy hormones from each donor, or address/discuss the possibility of differences mediated by those changes in age range.

This is a good point but unfortunately we have not collected at the time of recruitment the info on menopausal status as well as on hormone replacement therapies. We apologize.

2. It would be appropriate to include whether donors with type 2 diabetes have been excluded, to avoid anti-inflammatory effects of glycemic control drugs such as metformin, considering they are all in the obese category and there are chances of donors being either pre-diabetic or diabetic.

Individuals with T2D are the majority of participants in both groups, and all take Metformin. We do not have participants with T2D not taking this drug.

3. Include in the discussion whether these changes are predicted or not in males.

Yes, these changes are expected to occur also in male individuals. We have very preliminary data on male donors giving both breast and pannus AT, in which the pannus is more inflammatory than the breast.

A comment on this point has been added to Conclusions.

4. Add in the discussion if there are any known changes in the frequency of DN B cells between males and females (mice and/or humans).

In male donors (see above reply), frequencies of DN B cells are comparable to those found in female donors.

Also this point is in Conclusions.

5. Extend the discussed part about the potential therapeutic effect of depleting B cells versus depleting just DN B cells.

Thank you for this suggestion. A sentence has been added at the end of Conclusions.

Response to editorial office

We have also checked the manuscript for the occurrence of overlapping text with our previous publications. We apologize for this.

---

## [Decision Letter · Decision Letter 1]

24 Jan 2023

PONE-D-22-22884R1Phenotypic and functional features of B cells from two different human subcutaneous adipose depotsPLOS ONE

Dear Dr. Frasca,

Thank you for submitting your manuscript to PLOS ONE. After careful consideration, we feel that it has merit but does not fully meet PLOS ONE’s publication criteria as it currently stands. Therefore, we invite you to submit a revised version of the manuscript that addresses the points raised during the review process. Your revised manuscript was reviewed by the original referees. As you will see they both recognize that the revised paper has been improved, and reviewer #2 suggested acceptance as it is. However, reviewer#1 pointed out several issues that will need to be properly addressed by the authors. 

We look forward to receiving your revised manuscript.

Kind regards,

Makoto Kanzaki, Ph.D.

Academic Editor

PLOS ONE

Journal Requirements:

Reviewers' comments:

Reviewer's Responses to Questions

**Comments to the Author**

1. If the authors have adequately addressed your comments raised in a previous round of review and you feel that this manuscript is now acceptable for publication, you may indicate that here to bypass the “Comments to the Author” section, enter your conflict of interest statement in the “Confidential to Editor” section, and submit your "Accept" recommendation.

Reviewer #1: (No Response)

Reviewer #2: All comments have been addressed

2. Is the manuscript technically sound, and do the data support the conclusions?

Reviewer #1: No

Reviewer #2: Yes

3. Has the statistical analysis been performed appropriately and rigorously? 

Reviewer #1: I Don't Know

Reviewer #2: Yes

4. Have the authors made all data underlying the findings in their manuscript fully available?

Reviewer #1: No

Reviewer #2: Yes

5. Is the manuscript presented in an intelligible fashion and written in standard English?

Reviewer #1: Yes

Reviewer #2: Yes

6. Review Comments to the Author

Reviewer #1: The authors have made many edits in response to my initial review. Unfortunately, these changes did not resolve the questions to my satisfaction, many concerns remain. Here is a summary of my concerns.

First, authors have added a statement in two different places in the Results and Introduction in this version which notes “…due to the ongoing process of class switch OCCURRING IN THE OBESE ADIPOSE TISSUE” (page 8) and “ongoing process of class switch occurring in the obese AT” (page 9). This is a very interesting point and one of great interest in the field. However, the paper the authors cite (their own) does not show this result, so this is an overstatement. The authors previously found CXCR5+ T cells and class-switched B cells are detectable in the SVF from the human adipose. This DOES NOT confirm that the class switching is happening in the adipose itself. This is a very important point, so it undermines the author’s credibility to over-interpret and over-state their own results. These statements need to be revised or removed.

There is a problem with the author’s solution to the reviewer concern in Figure 2. The authors have redrawn the gate in their top two panels to include only the CD21low cells as requested. This is now better. However, the actual percentages of cells provided for all the panels, including the 4 panels below which should have been sub-gated on the first panels are completely unchanged. This includes the percentages shown for all populations as well as the error ranges. It is not likely that altering the gate in the top panel to include fewer cells will leave all subsequent values and errors as EXACTLY the same. Please review your data analysis and update the presentation of the data to include only the cells in the newly drawn subgate. Given all the mistakes with the data so far, it will be essential to correct these flow plots and include a scatterplot of RNA expression in ALL 5 samples/treatment group plus the statistics so we can be reassured that this is a significant and meaningful change of Tbet expression in the CD21loCD95+ population across all the samples.

In Figure 3, again the authors have responded to the reviewer’s comments by reanalyzing the data and providing a new plot for the CD21 flow staining. The CD21 histogram now includes X-axis labels, which is helpful. However, the MFI of the isotype control sample is still almost double the MFI of the two experimental samples (Breast and pannus). This presents the same exact problem that existed with the first set of data- the isotype control provides a higher MFI than either of the experimental samples. This indicates that neither experimental sample has any positive expression of CD21 above background staining with an irrelevant antibody. It also means that these two samples cannot be compared to each other to make meaningful statements about their relative expression of “CD21” since they need to be considered negative for CD21 expression. Since this difference is the goal of the scatterplots, they need to be removed or the conclusion restricted to CD95 expression.

It appears that Fig 6 is still labeled as 2-NBDG…I am guessing this means the previous label was correct, but in the first draft, the actual data inside the panel was inadvertently swapped with the ROS data in Fig 9?

Labels and data presentation for Figure 9C looks corrected, but it would help to add a phrase in the results to state that the data in panel 9C includes data from both breast and pannus tissue, and it would be very helpful to identify which samples are breast vs pannus by using two colors or symbols in panel 9C. This should make the author’s conclusion even more apparent as the two groups should segregate differently.

Reviewer #2: Author have address as their best all comments. There are some limitation that commonly occur when working with human samples that i understand cannot be fixed. For the most part, conclusions in the article are rigorously supported by data shown.

7. PLOS authors have the option to publish the peer review history of their article (what does this mean?). If published, this will include your full peer review and any attached files.

Reviewer #1: No

Reviewer #2: No

---

## [Author Response · Author response to Decision Letter 1]

15 Feb 2023

Reviewer #1: The authors have made many edits in response to my initial review. Unfortunately, these changes did not resolve the questions to my satisfaction, many concerns remain. Here is a summary of my concerns.

Thank you very much! Below (in red) is our response to your additional comments. Changes in the second amended version of our manuscript are highlighted in yellow.

First, authors have added a statement in two different places in the Results and Introduction in this version which notes “…due to the ongoing process of class switch OCCURRING IN THE OBESE ADIPOSE TISSUE” (page 8) and “ongoing process of class switch occurring in the obese AT” (page 9). This is a very interesting point and one of great interest in the field. However, the paper the authors cite (their own) does not show this result, so this is an overstatement. The authors previously found CXCR5+ T cells and class-switched B cells are detectable in the SVF from the human adipose. This DOES NOT confirm that the class switching is happening in the adipose itself. This is a very important point, so it undermines the author’s credibility to over-interpret and over-state their own results. These statements need to be revised or removed.

Good point, thank you! We added a paragraph in the Introduction (we believe that it is better there than in Results and Discussion) to explain this point – we hope in a clarifying way for the reviewer and the readers. 

In addition to the published data, we have evidence from 3 donors only of post-switch transcripts (Iμ-Cγ1) in B cells sorted from the SVF, measured by semiquantitative PCR. Therefore, we are confident that the obese AT is a site of ongoing class switch recombination. 

There is a problem with the author’s solution to the reviewer concern in Figure 2. The authors have redrawn the gate in their top two panels to include only the CD21low cells as requested. This is now better. However, the actual percentages of cells provided for all the panels, including the 4 panels below which should have been sub-gated on the first panels are completely unchanged. This includes the percentages shown for all populations as well as the error ranges. It is not likely that altering the gate in the top panel to include fewer cells will leave all subsequent values and errors as EXACTLY the same. Please review your data analysis and update the presentation of the data to include only the cells in the newly drawn subgate. Given all the mistakes with the data so far, it will be essential to correct these flow plots and include a scatterplot of RNA expression in ALL 5 samples/treatment group plus the statistics so we can be reassured that this is a significant and meaningful change of Tbet expression in the CD21loCD95+ population across all the samples.

Another good point, thank you for pointing this to us in the first review. Please note that the frequencies of CD21lowCD95+ B cells were correctly calculated from the beginning (now checked again) even when the CD21 gate was not correct in the figure. I don’t know what happened with that gate when we first submitted Fig. 2. My apologies!

In Figure 3, again the authors have responded to the reviewer’s comments by reanalyzing the data and providing a new plot for the CD21 flow staining. The CD21 histogram now includes X-axis labels, which is helpful. However, the MFI of the isotype control sample is still almost double the MFI of the two experimental samples (Breast and pannus). This presents the same exact problem that existed with the first set of data- the isotype control provides a higher MFI than either of the experimental samples. This indicates that neither experimental sample has any positive expression of CD21 above background staining with an irrelevant antibody. It also means that these two samples cannot be compared to each other to make meaningful statements about their relative expression of “CD21” since they need to be considered negative for CD21 expression. Since this difference is the goal of the scatterplots, they need to be removed or the conclusion restricted to CD95 expression.

Autoimmune B cells are CD21low/negCD95+, therefore removing CD21 staining is not good if we want to show that these pathogenic B cells are expanded in the obese AT.

Unfortunately, the isotype control antibody for fluorochrome-conjugated CD21 (recommended by the vendor) has always given issues to us - but the MFI graph even with this isotype control antibody shows less CD21 expression in pannus as compared to breast samples. We have thawed the SVFs and re-stained them using another isotype control (a bit better than the previous one). A new Fig 3 is provided with this second amended submission.

It appears that Fig 6 is still labeled as 2-NBDG…I am guessing this means the previous label was correct, but in the first draft, the actual data inside the panel was inadvertently swapped with the ROS data in Fig 9?

Correct! Fig. 6 shows 2-NBDG results whereas Fig. 9 ROS results. The previous mistake has been fixed in the first amended version.

Labels and data presentation for Figure 9C looks corrected, but it would help to add a phrase in the results to state that the data in panel 9C includes data from both breast and pannus tissue, and it would be very helpful to identify which samples are breast vs pannus by using two colors or symbols in panel 9C. This should make the author’s conclusion even more apparent as the two groups should segregate differently.

This was not clarified because in Fig. 9C, as well as in every other figure of the paper, breast samples are identified with round symbols whereas pannus samples are identified with square symbols. 

Reviewer #2: Author have address as their best all comments. There are some limitation that commonly occur when working with human samples that i understand cannot be fixed. For the most part, conclusions in the article are rigorously supported by data shown.

Thank you very much! We really appreciate this comment.

---

## [Decision Letter · Decision Letter 2]

23 Mar 2023

PONE-D-22-22884R2Phenotypic and functional features of B cells from two different human subcutaneous adipose depotsPLOS ONE

Dear Dr. Frasca,

Thank you for submitting your manuscript to PLOS ONE. After careful consideration, we feel that it has merit but does not fully meet PLOS ONE’s publication criteria as it currently stands. Therefore, we invite you to submit a revised version of the manuscript that addresses the points raised during the review process. Your revised manuscript was reviewed by the original referees, and a reviewer still has concerns that will need to be properly addressed by the authors. The authors need to respond to his/her constructive comments.  Please submit your revised manuscript by May 07 2023 11:59PM. If you will need more time than this to complete your revisions, please reply to this message or contact the journal office at plosone@plos.org. Please include the following items when submitting your revised manuscript:A rebuttal letter that responds to each point raised by the academic editor and reviewer(s). You should upload this letter as a separate file labeled 'Response to Reviewers'.A marked-up copy of your manuscript that highlights changes made to the original version. You should upload this as a separate file labeled 'Revised Manuscript with Track Changes'.An unmarked version of your revised paper without tracked changes. You should upload this as a separate file labeled 'Manuscript'.If applicable, we recommend that you deposit your laboratory protocols in protocols.io to enhance the reproducibility of your results. Protocols.io assigns your protocol its own identifier (DOI) so that it can be cited independently in the future. For instructions see: https://journals.plos.org/plosone/s/submission-guidelines#loc-laboratory-protocols. Additionally, PLOS ONE offers an option for publishing peer-reviewed Lab Protocol articles, which describe protocols hosted on protocols.io. Read more information on sharing protocols at https://plos.org/protocols?utm_medium=editorial-email&utm_source=authorletters&utm_campaign=protocols.

We look forward to receiving your revised manuscript.

Kind regards,

Makoto Kanzaki, Ph.D.

Academic Editor

PLOS ONE

Journal Requirements:

Reviewers' comments:

Reviewer's Responses to Questions

**Comments to the Author**

1. If the authors have adequately addressed your comments raised in a previous round of review and you feel that this manuscript is now acceptable for publication, you may indicate that here to bypass the “Comments to the Author” section, enter your conflict of interest statement in the “Confidential to Editor” section, and submit your "Accept" recommendation.

Reviewer #1: All comments have been addressed

Reviewer #2: (No Response)

2. Is the manuscript technically sound, and do the data support the conclusions?

Reviewer #1: Yes

Reviewer #2: Partly

3. Has the statistical analysis been performed appropriately and rigorously? 

Reviewer #1: Yes

Reviewer #2: Yes

4. Have the authors made all data underlying the findings in their manuscript fully available?

Reviewer #1: Yes

Reviewer #2: Yes

5. Is the manuscript presented in an intelligible fashion and written in standard English?

Reviewer #1: Yes

Reviewer #2: Yes

6. Review Comments to the Author

Reviewer #1: (No Response)

Reviewer #2: 1. Figure 8C shows ECAR measurements over time. Usually, when running mito stress kits for seahorse, when ECAR data is generated do not show the same pattern that OCR. It seems like Oligomycin (time point 20min) had no effect on ECAR, however, it is expected to increase ECAR levels, as cells tend to compensate the inhibition of ATP produced by the mitochondria. Additionally, FCCP and Rot/AA do not have an effect in ECAR . Because FCCP promotes maximal respiration, that should not reduce acidification rate. Another thing worth checking is the values on the Y axis for ECAR. Normally, they are (at least the basal) within the range of the OCR value. I encourage authors to review raw data to make sure the data shown is correct or explain why this might have happened.

2. When ROS were measured, were cells fixed? I was concerned about the incubation times as usually ROS production in live cells is measured within 20-30min. As it is described in material and methods, ROS were measured ~1h after the dye was added. If authors have some kind of positive control for ROS, like H2O2 treatment will help clarifying this.

When referring to ROS, I suggest using the word "production" rather than "ROS expression"

3. Figure 7 shows increased LDH expression in pannus samples. Authors suggest that this increase is due to pannus being more hypoxic than breast environment. This statement is an overinterpretation, since usually in vitro culture conditions do not mimic hypoxic conditions in the body and also because PDH mRNA levels are also upregulated. I would suggest a more detailed and careful explanation/correlation. Moreover, the fact that both, OCR and ECAR are higher in pannus cells, will implicate that these cells are more metabolically active than cells from breast tissue, but does not necessarily mean these cells are more glycolytic and/or anaerobic. Authors should extend the discussion on why these changes in metabolism could be occurring once ECAR data is reviewed.

7. PLOS authors have the option to publish the peer review history of their article (what does this mean?). If published, this will include your full peer review and any attached files.

Reviewer #1: No

Reviewer #2: No

---

## [Author Response · Author response to Decision Letter 2]

28 Mar 2023

Reviewer #2: 

Dear reviewer, below (in red) is our response to your comments. Changes in the third amended version of our manuscript are highlighted in light blue.

1. Figure 8C shows ECAR measurements over time. Usually, when running mito stress kits for seahorse, when ECAR data is generated do not show the same pattern that OCR. It seems like Oligomycin (time point 20min) had no effect on ECAR, however, it is expected to increase ECAR levels, as cells tend to compensate the inhibition of ATP produced by the mitochondria. Additionally, FCCP and Rot/AA do not have an effect in ECAR. Because FCCP promotes maximal respiration, that should not reduce acidification rate. Another thing worth checking is the values on the Y axis for ECAR. Normally, they are (at least the basal) within the range of the OCR value. I encourage authors to review raw data to make sure the data shown is correct or explain why this might have happened.

We have checked our ECAR results as well as results from other groups in the literature. We can say that ECAR profiles may show either a small effect of oligomycin (if any) as well as a reduction of ECAR values after injection of Rot/AA (as an example, please see refs from Dequina et al. (DOI: 10.1016/j.cmet.2019.07.004 and DOI: 10.1371/journal.pone.0170975). 

To further support our results in Fig. 8, we would like to tell the reviewer that we have also run glycolytic tests in Seahorse injecting Glucose, oligomycin and 2-DG, and these tests have confirmed a highly glycolytic phenotype of B cells from the pannus as compared to those from the breast. Please see these results below (pannus=squared symbols, breast=round symbols).

The figure is attached to the uploaded response to reviewers.

2. When ROS were measured, were cells fixed? I was concerned about the incubation times as usually ROS production in live cells is measured within 20-30min. As it is described in material and methods, ROS were measured ~1h after the dye was added. If authors have some kind of positive control for ROS, like H2O2 treatment will help clarifying this.

As to the first point, cells were not fixed, they were acquired immediately at the end of staining (as already indicated in Materials and methods, paragraph “Detection of Reactive Oxygen Species (ROS) using CellROX Oxidative Stress Reagent”).

When referring to ROS, I suggest using the word "production" rather than "ROS expression".

Thank you for the suggestion. We changed the text accordingly (pages 13-14).

3. Figure 7 shows increased LDH expression in pannus samples. Authors suggest that this increase is due to pannus being more hypoxic than breast environment. This statement is an overinterpretation, since usually in vitro culture conditions do not mimic hypoxic conditions in the body and also because PDH mRNA levels are also upregulated. I would suggest a more detailed and careful explanation/correlation. Moreover, the fact that both, OCR and ECAR are higher in pannus cells, will implicate that these cells are more metabolically active than cells from breast tissue, but does not necessarily mean these cells are more glycolytic and/or anaerobic. Authors should extend the discussion on why these changes in metabolism could be occurring once ECAR data is reviewed.

Thank you also for this suggestion. We changed the text accordingly (pages 12-13).

---

## [Decision Letter · Decision Letter 3]

14 Apr 2023

Phenotypic and functional features of B cells from two different human subcutaneous adipose depots

PONE-D-22-22884R3

Dear Dr. Frasca,

We’re pleased to inform you that your manuscript has been judged scientifically suitable for publication and will be formally accepted for publication once it meets all outstanding technical requirements.

Kind regards,

Makoto Kanzaki, Ph.D.

Academic Editor

PLOS ONE

Additional Editor Comments (optional):

Reviewers' comments:

Reviewer's Responses to Questions

**Comments to the Author**

1. If the authors have adequately addressed your comments raised in a previous round of review and you feel that this manuscript is now acceptable for publication, you may indicate that here to bypass the “Comments to the Author” section, enter your conflict of interest statement in the “Confidential to Editor” section, and submit your "Accept" recommendation.

Reviewer #2: All comments have been addressed

2. Is the manuscript technically sound, and do the data support the conclusions?

Reviewer #2: Yes

3. Has the statistical analysis been performed appropriately and rigorously? 

Reviewer #2: Yes

4. Have the authors made all data underlying the findings in their manuscript fully available?

Reviewer #2: Yes

5. Is the manuscript presented in an intelligible fashion and written in standard English?

Reviewer #2: Yes

6. Review Comments to the Author

Reviewer #2: Authors have successfully addressed all reviewer 2 comments. However, the indicated references to address comment#1 were not very convincing. reviewr 2 strongly suggest to add the glycolitic assay authors show in the comment response as main figure along with the other seahorse results.

7. PLOS authors have the option to publish the peer review history of their article (what does this mean?). If published, this will include your full peer review and any attached files.

Reviewer #2: No

---

## [Editor Report · Acceptance letter]

19 Apr 2023

PONE-D-22-22884R3 

Phenotypic and functional features of B cells from two different human subcutaneous adipose depots 

Dear Dr. Frasca:

I'm pleased to inform you that your manuscript has been deemed suitable for publication in PLOS ONE. Congratulations! Your manuscript is now with our production department. 

Kind regards, 

on behalf of

Dr. Makoto Kanzaki 

Academic Editor

PLOS ONE